# Learn to Select Node in Branch and Bound with Causality Modeling

## Abstract

Learning-based approaches have shown strong promise in branch and bound (BnB) node selection by using offline data. They typically model *correlations* from node features to node quality, selecting nodes based on predicted quality. However, this correlation modeling may encode spurious patterns rather than the true decision rationale. For example, it may associate node quality with lower bounds; but a node with lower bounds is not necessarily better due to overestimated relaxation, illustrating how this feature-level signal misleads decision. The true decision rationale lies in whether a node contains an optimal solution. To this, this paper proposes modeling the *causal* effect of optimal solution presence on node selection, moving beyond correlation modeling. We define the causal signal by BnB's optimality transitivity: if a node contains an optimal solution, then its parent must also contain that solution; consequently, optimal nodes tend to resemble their parents in feature representation. We implement this by contrastive learning, treating parent-child node pairs in which both nodes contain an optimal solution as positive samples and other pairs as negative; training the model to distinguish nodes containing an optimal solution from those that do not. This enables learning intrinsic node representations centered on optimality, free from spurious correlations. Experiments show that our method significantly outperforms correlation-based approaches in efficiency, robustness, and generalization, achieving near-expert performance under limited data and distribution shift.

## 1 Introduction

Mixed integer linear programming (MILP) is a fundamental formulation for NP-hard optimization (Floudas & Lin 2005, Ren & Gao 2010, Paschos 2014). Branch and bound (BnB) (Land & Doig 1960) is a dominant algorithmic framework for MILPs. It conducts a tree-based search that recursively decomposes the original problem (root) into subproblems (child nodes) and prunes suboptimal regions. Node selection is crucial in BnB, as it shapes the search trajectory and affects efficiency by prioritizing promising subproblems (Achterberg 2007).

Traditional BnB node selection approaches are heuristic-based, relying on problem features like lower bounds and feasibility (Achterberg et al. 2008). They are fast and easy to implement, but may lead to suboptimal search trajectories due to greedy decisions and lack generalization due to the reliance on specific features. Learning-based approaches, e.g., imitation learning (He et al. 2014, Song et al. 2018, Labassi et al. 2022) and reinforcement learning (Mattick & Mutschler 2024, Zhang et al. 2025), aim to address these issues by learning expert trajectories or environment feedback from a distribution of target problem instances (Bengio et al. 2021). They are particularly effective when a priori computational resources allow offline training, which in turn enables online inference on many problem instances from the target domain.

Learning-based node selection approaches usually model statistical correlations from node features to node quality, where quality is defined by solver-provided signals such as linear programming (LP) bounds, primal-dual gap, or oracle selection labels. The models then select nodes based on predicted quality scores[1]. However, a key challenge is that insufficient diversity in training data can

---

[1]While the input to the model is not necessarily predefined node features (which may be a structural representation like a graph), the model implicitly extracts and map features from the input to node quality scores.

cause models to capture spurious patterns rather than the true decision rationale. Although prior work attempts to mitigate this by sampling from diverse trajectories and wide trees, the risk remains significant, as it is difficult to fully control data diversity and representativeness. For example, (i) models may favor nodes with smaller lower bounds, which are frequently expanded in solver runs, though this can be misleading due to overestimated relaxations; (ii) models can inherit depth bias when trajectories are skewed toward depth-first rollouts, leading to premature descent into suboptimal branches; and (iii) models may prefer certain branching variables or feasible relaxations, which often correlate with high-quality nodes but mostly reflect instance-specific heuristics.

The true decision rationale actually lies in whether a node contains an optimal solution. This calls for a shift from correlation-based to causality-based modeling, i.e., a node should be selected because it contains an optimal solution. To this, this paper proposes explicitly modeling the causal effect of optimal solution presence on node selection. Our contributions are:

- We propose causality modeling for BnB node selection. Through rigorous analysis, we demonstrate that correlation-based modeling is theoretically fragile. We instead propose the causality modeling that captures the true decision rationale of whether a node contains an optimal solution.

- We define BnB's optimality transitivity as the causal signal. In BnB, optimality is transitive: if a node contains an optimal solution, then its parent must also contain that solution. Consequently, a node containing an optimal solution tends to exhibit feature representation that resembles that of its parent. This signal is invariant to depth, search order, and instance-specific features, providing a principled foundation for identifying optimal nodes.

- We implement the causality modeling with contrastive learning. We treat parent-child node pairs containing optimal solutions as positive samples, and all others as negative. The model is trained to distinguish nodes containing optimal solutions from those that do not. This aligns node selection with the true decision rationale, i.e., containing an optimal solution.

- A data augmentation strategy further enhances generalization, enabling data augmentation with limited expert intervention.

We validate the proposed approach on three MILP benchmarks and two real-world datasets. Experiments show that it achieves much shorter solve times and smaller search trees than correlation-based baselines. On the real-world datasets, it solves the most instances with the highest solution quality. Moreover, the data augmentation ensures strong performance under limited expert supervision.

The rest of the paper includes: Section 2 introduces background and related work. Sections 3 presents the proposed causality modeling. Section 4 reports experiments. Section 5 concludes.

## 2 BACKGROUND AND RELATED WORK

### 2.1 MIXED INTEGER LINEAR PROGRAMMING

MILP refers to an optimization problem with a linear objective and linear constraints, where some decision variables are constrained to be integers and others are continuous. The standard form is

$$
\begin{aligned}
\min \quad & \mathbf{c}^T \mathbf{x}, \\
\text{s.t.} \quad & \mathbf{A}\mathbf{x} \le \mathbf{b}, \quad x_i \in \mathbb{Z}, i \in I, \quad x_j \in \mathbb{R}, j \notin I.
\end{aligned}
\tag{1}
$$

where $\mathbf{x} \in \mathbb{R}^n$ denotes decision variables, $i \in I$ is the index set of integer-constrained variables, $\mathbf{c} \in \mathbb{R}^n$ is the objective coefficient vector, and $\mathbf{A} \in \mathbb{R}^{m \times n}$, $\mathbf{b} \in \mathbb{R}^m$ define the linear constraints.

### 2.2 BRANCH AND BOUND AND ITS NODE SELECTION

BnB (Land & Doig 1960) is a tree-based framework for solving MILPs. It recursively decomposes the problem into subproblems, with each node representing a subproblem defined by additional bound constraints. Branching creates child nodes by imposing disjoint bounds on a fractional variable, and the algorithm proceeds by branching, node selection, and pruning.

Traditional BnB node selection approaches are heuristic-based, relying on handcrafted rules derived from problem features. Common strategies include best estimate search (Bénichou et al. 1971, Forrest et al. 1974), best first search (Hart et al. 1968, Achterberg 2007), and depth first search (Dakin 1965). While these rules are simple and computationally efficient, they often yield suboptimal trajectories due to their greedy nature and generalize poorly across problem instances.

Learning-based approaches aim to overcome these limitations by training node selection policies from expert trajectories (imitation learning (He et al. 2014, Song et al. 2018, Labassi et al. 2022)) or environment feedback (reinforcement learning (Zhang et al. 2025, Mattick & Mutschler 2024)). In imitation learning, He et al. (2014) used DAGGER (Ross et al. 2011) to train a support vector machine to mimic a diving oracle. The oracle selects nodes based on a known optimal solution, guiding search while avoiding detours. Song et al. (2018) proposed retrospective imitation, using a RankNet (Burges et al. 2005) to follow the shortest path to the optimum. Their method reduces gaps under fixed node limits. Building on Gasse et al. (2019), Labassi et al. (2022) used a graph neural network (GNN) for pairwise node comparison. They incorporated features like node depth and global bounds with the GNN, which reduces nodes explorations, though solving time still lags behind heuristics. Recent work has also explored reinforcement learning , showing that effective policies can be learned without relying on expert demonstrations (Zhang et al. 2025, Mattick & Mutschler 2024).

Related work on contrastive learning is given in Section A.1 of the Appendix.

## 3 CAUSALITY MODELING FOR BnB NODE SELECTION

### 3.1 THEORETICAL MOTIVATION

Existing correlation-based BnB node selection approaches are theoretically fragile. They treat BnB node selection as *correlation* estimation by modeling the *observational* distribution

$$\mathbb{P}(y \mid u_t = v,\, s_t), \tag{2}$$

where $s_t$ is the search tree at step $t$, $u_t$ is the action of selecting candidate node $v$, and $y$ is a binary or scalar proxy that directly or implicitly indicates whether expanding $v$ lies on an optimal path. By the law of total probability,

$$\mathbb{P}(y \mid u_t = v,\, s_t) = \sum_z \mathbb{P}(y \mid u_t = v,\, s_t,\, z)\, \mathbb{P}(z \mid u_t = v,\, s_t), \tag{3}$$

where $z$ denotes latent confounders such as node depth, LP bounds, or branching heuristics. In practice, nodes that contain the optimal solution often share particular characteristics(e.g., those with small LP bounds, which are frequently expanded in solver runs). These characteristics arise because the B&B process progressively restricts the feasible region along the search path, causing optimal nodes to appear in parts of the tree where such confounder-related features are naturally present. Consequently, observational data obtained from solver runs inherently couples optimality with these confounders. When a correlation-based model is trained on this data, the estimated relationship $P(y \mid u_t = v, s_t)$ can be dominated by these spurious associations, leading the model to rely on confounder-driven features rather than on information that truly reflects whether a node leads to the global optimum.

In contrast, node selection is inherently a *causal* decision-making problem, which corresponds to estimating the *interventional* effect under Pearl's *do*-calculus (Pearl 2009, Hernán & Robins 2020)

$$\mathbb{P}\big(y \mid do(u_t = v),\, s_t\big), \tag{4}$$

where $do(u_t = v)$ denotes an external intervention that enforces the expansion of node $v$ at $s_t$. This interventional distribution characterizes the outcome $y$ independently of the original search policy or latent confounders. When the standard identifiability conditions (i.e., consistency, back-door admissibility, and positivity) (Pearl 2009, Hernán & Robins 2020) hold with confounders $z$, the causal estimand admits the back-door formula

$$\mathbb{P}\big(y \mid do(u_t = v),\, s_t\big) = \sum_z \mathbb{P}(y \mid u_t = v,\, s_t,\, z)\, \mathbb{P}(z \mid s_t). \tag{5}$$

The back-door adjustment controls for $z$ by reweighting them according to their marginal distribution $\mathbb{P}(z \mid s_t)$, rather than the biased observational distribution $\mathbb{P}(z \mid u_t = v, s_t)$ that arises when data are sampled conditional on $u_t = v$.

The estimation error incurred when using the biased observational target instead of the intended interventional target is

$$
\begin{aligned}
\Delta(v, s_t) &:= \mathbb{P}\big(y \mid do(u_t = v), s_t\big) - \mathbb{P}\big(y \mid u_t = v, s_t\big) \\
&= \sum_z \mathbb{P}(y \mid u_t = v, s_t, z)\big(\mathbb{P}(z \mid s_t) - \mathbb{P}(z \mid u_t = v, s_t)\big),
\end{aligned}
\tag{6}
$$

which yields the bound

$$
\big|\Delta(v, s_t)\big| \;\leq\; 2\,\mathrm{TV}(\mathbb{P}(z \mid s_t),\, \mathbb{P}(z \mid u_t = v, s_t)),
\tag{7}
$$

where TV is the total variation distance (Kallus & Zhou 2021). It can be seen that the confounding gap scales with the sampling bias between $\mathbb{P}(z \mid s_t)$ and $\mathbb{P}(z \mid u_t = v, s_t)$. As BnB is sequential, the bias is amplified over time. Letting $R(v)$ be the expected remaining expansions to reach an optimum from node $v$, $v_t^{\pi_\theta}$ and $v_t^{\pi^*}$ represent the nodes selected at step $t$ under the learned policy $\pi_\theta$ and the optimal policy $\pi^*$, respectively, the instantaneous regret $r_t = \mathbb{E}[R(v_t^{\pi_\theta}) - R(v_t^{\pi^*})]$ satisfies

$$
\sum_{t=1}^{T} r_t \;\geq\; \sum_{t=1}^{T} \Pr(v_t^{\pi_\theta} \neq v_t^{\pi^*})\, \alpha_t,
\tag{8}
$$

where $\alpha_t$ denotes a lower bound on the expected conditional detour when deviating from the optimal branch at step $t$. Consequently, even small error probabilities accumulate linearly in regret, and, in the worst case, may lead to exponential growth in search cost due to exploration of suboptimal subtrees of exponential size.

These observations motivate a causality modeling that estimates $\mathbb{P}(y \mid do(u_t = v), s_t)$. In essence, correlation-based modeling reduces node selection to pattern recognition, while causality modeling frames it as decision-making, which faithfully captures the causal mechanism of BnB search.

### 3.2 Proposed Causality Modeling

We aim to model the causal effect of optimal-solution presence on node selection. While our model does not assume strict satisfaction of the back-door conditions in practice (since not all confounders $z$ may be explicitly observed or controlled), the back-door formula provides an important theoretical principle, i.e., the causal target $\mathbb{P}(y \mid do(u_t = v), s_t)$ should be obtained independently of the biased observational distribution $\mathbb{P}(z \mid u_t = v, s_t)$. Motivated by this, we leverage a structural property of BnB, namely *optimality transitivity*: if a node contains an optimal solution, all its ancestors must also contain that solution. This transitivity leaves structural traces of optimal solutions and provides a principled supervision signal for causal modeling.

Formally, let $\mathbf{x}^*$ be an optimal solution of a MILP instance, and $\mathcal{F}_v$ the feasible region of node $v$, defined as $\mathcal{F}_v := \{\mathbf{x} \in \mathbb{R}^n \mid \mathbf{A}\mathbf{x} \leq \mathbf{b},\, \mathbf{x} \text{ satisfies all branching constraints from root to } v\}$. We have:

**Theorem 1** (Optimality Transitivity). *If* $\mathbf{x}^* \in \mathcal{F}_v$*, we have* $\mathbf{x}^* \in \mathcal{F}_{v^{\mathrm{anc}}}$ *for any ancestor* $v^{\mathrm{anc}}$ *of* $v$.

*Proof.* Each branching step appends additional bound constraints, implying that feasible regions shrink monotonically along the tree: $\mathcal{F}_v \subseteq \cdots \subseteq \mathcal{F}_{\mathrm{root}}$. Hence, if $\mathbf{x}^* \in \mathcal{F}_v$, $\mathbf{x}^* \in \mathcal{F}_{v^{\mathrm{anc}}}$. □

This theorem establishes a causal linkage between a node and its ancestor along an optimal path. To implement this, we model node-ancestor similarity as a proxy for the interventional likelihood of optimality. Specifically, we require a node on an optimal path to share similar feature representation with its ancestor. Let $\phi(v)$ and $\phi(v^{\mathrm{anc}})$ denote the embeddings of node $v$ and its ancestor $v^{\mathrm{anc}}$, encoded by a shared function $\phi(\cdot)$. We define a similarity measure

$$
H : \big(\phi(v), \phi(v^{\mathrm{anc}})\big) \mapsto \mathbb{R},
\tag{9}
$$

where $H$ can be instantiated by inner product $\phi(v)^\top \phi(v^{\mathrm{anc}})$ or cosine similarity $\frac{\phi(v)^\top \phi(v^{\mathrm{anc}})}{\|\phi(v)\|\|\phi(v^{\mathrm{anc}})\|}$. Selecting candidates with maximal similarity favors nodes structurally aligned with an optimal path.

Crucially, the learned similarity aligns with the interventional distribution in equation 4. Compared with correlation-based modeling, the proposed causal similarity has two key advantages. First, by grounding node selection on the structural transitivity of optimality, the supervision depends solely on ancestor–child relations. This provides a principled way to reduce the confounding gap in equation 6 without directly accessing $z$. Second, the causal supervision improves out-of-distribution generalization. Even when test instances differ in size or constraint distribution, the transitivity principle remains valid, providing a stable structural prior.

### 3.3 IMPLEMENT CAUSALITY MODELING WITH CONTRASTIVE LEARNING

Contrastive learning is well suited for learning similarity functions, making it a natural choice for learning $H$ in equation 9. In our setting, a node and its ancestor form a *positive* pair if both contain an optimal solution, and a *negative* pair otherwise. The model is trained to assign higher similarity to pairs on optimal paths. This ensures that the learned similarity reflects the likelihood of containing an optimal solution.

#### 3.3.1 DATA COLLECTION AND AUGMENTATION

Like other learning-based node selection models, our model is trained offline on a distribution of target problem instances and deployed online to solve new instances from the target domain. Thus, we begin by collecting training data. Details are given in Section A.2.1. Specifically, data collection requires solving each MILP twice, i.e., first to obtain optimal solutions, and then to simulate a guided BnB traversal to identify nodes containing the optimal solutions. To reduce this overhead, we introduce a data augmentation strategy that avoids the second traversal by reusing information from the first solve. Details are given in Section A.2.2.

#### 3.3.2 TRAINING

To align the nodes with the causal signal of optimality, we adopt a contrastive learning objective that maximizes the mutual information between node embeddings and their causal signals in the BnB process. Each node is encoded by a bipartite graph encoder $\phi_\theta$ to obtain its embedding $\mathbf{v}_{\text{avg}}$ (details in Section A.2.3). The similarity between two nodes $v_a$ and $v_b$ with graph representations $G_a$ and $G_b$ is then defined via cosine:

$$H(v_a, v_b) = \cos\_\text{sim}(\phi_\theta(G_a), \phi_\theta(G_b)) = \frac{\mathbf{v}_{\text{avg},a}^\top \mathbf{v}_{\text{avg},b}}{\|\mathbf{v}_{\text{avg},a}\| \, \|\mathbf{v}_{\text{avg},b}\|}. \tag{10}$$

We then employ InfoNCE (Oord et al. 2018) to learn $H$. From an information-theoretic perspective, InfoNCE serves as a lower bound on the mutual information between inputs and labels. In our context, this means learning to distinguish between nodes that do and do not contain an optimal solution. Let $\mathcal{D}$ be the set of training data constructed in Sections A.2.1 and A.2.2, the loss is defined as

$$\mathcal{L}(\theta) = \frac{-1}{|\mathcal{D}|} \sum_{(v^+, v^-, v^{\text{anc}}) \in \mathcal{D}} \log \frac{\exp\left(H(v^{\text{anc}}, v^+)/\tau\right)}{\exp\left(H(v^{\text{anc}}, v^+)/\tau\right) + \exp\left(H(v^{\text{anc}}, v^-)/\tau\right)}, \tag{11}$$

where $v^+$ is a positive candidate node that contains an optimal solution, $v^-$ is a negative candidate without this optimal solution, $v^{\text{anc}}$ is the lowest common ancestor of $v^+$ and $v^-$ in the BnB tree; $(v^+, v^-, v^{\text{anc}})$ form a data triple (details in Section A.2.1); $\tau$ is a temperature hyperparameter.

We provide a theoretical view of why the contrastive loss aligns with the causal target.

**Definition 1** (Causal estimand). *For a node $v$ and its ancestor $v^{anc}$, the target of node selection is to rank the candidate successors (children) of $v^{anc}$ by $P(y = 1 \mid v^{anc}, v)$.*

**Proposition 1** (Ranking consistency). *The Bayes-optimal solution of the InfoNCE loss is a monotone transform of $\log \frac{P(y=1|v^{anc},v)}{P(y=0|v^{anc},v)}$. Hence minimizing the loss recovers the correct ranking under mild assumptions (see Assumptions 1–2 in the Appendix).*

This shows that our contrastive objective is theoretically grounded. It aligns the learned score with the causal target. Full proofs, Fisher consistency, mutual-information analysis, and generalization bounds are provided in Section A.3 of the Appendix.

Table 1: Comparison of CausalM with baselines on the standard test data.

| Methods | FCMCNF | | | MAXSAT | | | GISP | | |
|---|---|---|---|---|---|---|---|---|---|
| | Time (s) | Nodes | Wins | Time (s) | Nodes | Wins | Time (s) | Nodes | Wins |
| Oracle | 3.38 ± 1.42 | 14.99 ± 4.13 | - | 5.28 ± 1.72 | 102.16 ± 2.48 | - | 3.68 ± 1.25 | 98.00 ± 3.29 | - |
| Estimate | 3.60 ± 1.48 | 21.40 ± 5.32 | 8 | 6.87 ± 1.69 | 176.58 ± 2.33 | 3 | 4.02 ± 1.25 | 218.06 ± 2.47 | 6 |
| SCIP | 4.04 ± 1.44 | 40.74 ± 4.57 | 11 | 7.99 ± 1.48 | 147.09 ± 1.96 | 2 | **3.82 ± 1.19** | 184.27 ± 1.96 | **23** |
| SVM | 3.49 ± 1.43 | 19.98 ± 5.05 | 7 | 6.23 ± 1.74 | 150.13 ± 2.60 | 3 | 3.99 ± 1.25 | **180.58 ± 2.78** | 2 |
| RankNet | 3.65 ± 1.48 | 20.96 ± 5.22 | 2 | 6.26 ± 1.75 | 142.13 ± 2.64 | 1 | 4.11 ± 1.28 | 203.49 ± 2.66 | 0 |
| L2C | 3.65 ± 1.50 | 21.24 ± 5.47 | 1 | 6.20 ± 1.90 | 138.03 ± 2.79 | 4 | 4.04 ± 1.27 | 184.31 ± 2.77 | 1 |
| CausalM | **3.43 ± 1.49** | **19.25 ± 5.07** | **21** | **5.28 ± 1.78** | **105.68 ± 2.50** | **37** | 3.82 ± 1.25 | 181.69 ± 2.77 | 18 |

Table 2: Comparison of CausalM with baselines on real-world datasets.

| Methods | CORLAT | | | MIK | | |
|---|---|---|---|---|---|---|
| | Time(s) | Nodes | Solved | Time(s) | Nodes | Solved |
| Oracle | 3.20 ± 2.64 | 58.46 ± 16.66 | 50 | 11.70 ± 1.53 | 362.27 ± 19.89 | 18 |
| Estimate | 4.86 ± 4.04 | **98.05 ± 28.59** | 48 | **13.44 ± 1.39** | 618.58 ± 12.63 | 18 |
| SCIP | **4.09 ± 3.48** | 109.47 ± 29.51 | **49** | 15.79 ± 1.59 | 629.40 ± 13.39 | 18 |
| SVM | 10.26 ± 7.53 | 188.32 ± 42.44 | 46 | 15.37 ± 1.58 | 597.06 ± 15.68 | 18 |
| RankNet | 9.60 ± 8.14 | 198.15 ± 51.20 | 43 | 17.01 ± 1.72 | 741.52 ± 16.05 | 18 |
| L2C | 25.15 ± 15.36 | 156.34 ± 33.50 | 42 | 22.36 ± 1.76 | 601.55 ± 13.94 | 18 |
| CausalM | 7.07 ± 4.32 | 120.82 ± 25.67 | **49** | 16.43 ± 1.46 | **581.89 ± 12.88** | 18 |

*Note.* Time(s) and Nodes are computed on the intersection of instances solved by all methods (i.e., the commonly solved subset).

## 4 EXPERIMENTS

### 4.1 SETUP

We evaluate the proposed causality modeling approach (termed as CausalM) on three widely-used NP-hard MILP benchmarks, FCMCNF (Hewitt et al. 2010), MAXSAT (Ansótegui & Gabàs 2017), GISP (Colombi et al. 2017), and two real-word datasets CORLAT (Gomes et al. 2008) and MIK (Atamtürk 2003). The proposed CausalM is compared with six baselibes, i.e., Oracle, Estimate, SCIP (Gamrath et al. 2020), SVM (He et al. 2014), RankNet (Song et al. 2018), and L2C (Labassi et al. 2022). Three performance metrics are used, i.e., solving time, number of expanded nodes, and comparison wins. Bold numbers in tabels denote the best results (except Oracle) according to pair-wise Wilcoxon sign test at 5% significance level. Details of the benchmarks, baselines, metrics, and experiment implementation are described in Section A.4. Code is available at `https://anonymous.4open.science/r/Causal_M-606F`.

### 4.2 IN-DISTRIBUTION RESULTS

We compare the proposed CausalM against baselines across the three benchmarks to evaluate its effectiveness and efficiency. The pairwise Wilcoxon signed-rank test (d Steel & Torrie 1986) is conducted to assess statistical significance of the results.

Table 10 reports results on the standard test sets. CausalM consistently ranks first or second across all three benchmarks, outperforming all learning-based and heuristic baselines. On FCMCNF and MAXSAT, it achieves the lowest average solve time and node count, along with the highest number of wins, demonstrating both efficiency and effectiveness. For the more challenging GISP, although CausalM does not surpass SCIP in overall performance, it closely matches it, i.e., ranking second in solving time and achieving a competitive win count. Notably, CausalM still outperforms all other learning-based methods on GISP. The performance gap between CausalM and the expert Oracle remains small, indicating that our method closely approximates expert-level node selection.

These results reflect the limitations of traditional heuristic-based approaches, which often rely on handcrafted rules such as lower bounds or search depth. Such heuristics, as exemplified by Estimate and SCIP, perform well when their assumptions match the problem structure, but degrade significantly in other cases. For example, while SCIP achieves strong results on GISP, its performance drops considerably on FCMCNF and MAXSAT, revealing poor generalization. Similarly, correlation-based learning methods such as RankNet and L2C exhibit consistently weak performance across benchmarks. In particular, RankNet fails to improve on structurally diverse problems

Table 3: Comparison of CausalM-aug with baselines on standard test data.

| Methods | FCMCNF | | | MAXSAT | | | GISP | | |
|---|---|---|---|---|---|---|---|---|---|
| | Time(s) | Nodes | Wins | Time(s) | Nodes | Wins | Time(s) | Nodes | Wins |
| Oracle | 3.38 ± 1.42 | 14.99 ± 4.13 | - | 5.28 ± 1.72 | 102.16 ± 2.48 | - | 3.68 ± 1.25 | 98.00 ± 3.29 | - |
| Estimate | 3.60 ± 1.48 | 21.40 ± 5.32 | 8 | 6.87 ± 1.69 | 176.58 ± 2.33 | 6 | 4.02 ± 1.25 | 218.06 ± 2.47 | 6 |
| SCIP | 4.04 ± 1.44 | 40.74 ± 4.57 | 11 | 7.99 ± 1.48 | 147.09 ± 1.96 | 3 | 3.82 ± 1.19 | 184.27 ± 1.96 | 18 |
| SVM | **3.49 ± 1.43** | **19.98 ± 5.05** | 9 | 6.23 ± 1.74 | 150.13 ± 2.60 | 12 | 3.99 ± 1.25 | **180.58 ± 2.78** | 3 |
| RankNet | 3.65 ± 1.48 | 20.96 ± 5.22 | 2 | 6.26 ± 1.75 | 142.13 ± 2.64 | 7 | 4.11 ± 1.28 | 203.49 ± 2.66 | 2 |
| L2C | 3.65 ± 1.50 | 21.24 ± 5.47 | 1 | **6.20 ± 1.90** | 138.03 ± 2.79 | **18** | 4.04 ± 1.27 | 184.31 ± 2.77 | 2 |
| CausalM-aug | 3.49 ± 1.52 | 21.23 ± 5.38 | **19** | 6.58 ± 1.79 | **132.94 ± 2.64** | 4 | **3.77 ± 1.24** | 206.67 ± 2.46 | **19** |

Table 4: Comparison of CausalM with baselines on transfer data.

| Methods | FCMCNF | | | MAXSAT | | | GISP | | |
|---|---|---|---|---|---|---|---|---|---|
| | Time(s) | Nodes | Wins | Time(s) | Nodes | Wins | Time(s) | Nodes | Wins |
| Oracle | 16.13 ± 1.72 | 74.59 ± 4.46 | - | 7.40 ± 1.49 | 160.33 ± 2.00 | - | 18.21 ± 1.53 | 1062.20 ± 1.81 | - |
| Estimate | 19.14 ± 1.93 | 122.37 ± 5.14 | 8 | 9.97 ± 1.55 | 247.01 ± 1.84 | 0 | 20.26 ± 1.68 | 1434.95 ± 1.84 | 1 |
| SCIP | 21.28 ± 1.84 | 178.45 ± 4.30 | 8 | 10.75 ± 1.34 | 170.92 ± 1.66 | 1 | **15.64 ± 1.48** | 1533.48 ± 1.85 | **37** |
| SVM | 18.30 ± 1.84 | 132.77 ± 4.73 | 12 | 8.80 ± 1.60 | 225.32 ± 2.06 | 2 | 18.53 ± 1.51 | 1221.79 ± 1.69 | 4 |
| RankNet | 19.76 ± 2.00 | **112.60 ± 5.50** | 3 | 9.50 ± 1.57 | 238.33 ± 2.11 | 1 | 18.89 ± 1.52 | 1215.05 ± 1.73 | 0 |
| L2C | 19.91 ± 1.90 | 120.17 ± 4.94 | 0 | 9.31 ± 1.70 | 233.10 ± 2.23 | 2 | 18.80 ± 1.52 | 1196.42 ± 1.75 | 0 |
| CausalM | **18.19 ± 1.84** | 114.16 ± 4.57 | **19** | **7.04 ± 1.49** | **155.93 ± 2.02** | **44** | 17.31 ± 1.51 | **1194.70 ± 1.72** | 8 |

like GISP, where depth, bounds, or relaxation signals no longer reliably indicate node quality. This suggests that it overfits to spurious feature patterns and fails to generalize under distribution shifts. In contrast, CausalM grounds its learning on the task-intrinsic causal signal of optimal solution presence. This design enables robust performance across all benchmarks. Additional results in Section A.5, including imitation accuracy and optimality gap, report the same observations and further support our analysis of CausalM's robust and generalizable performance.

As shown in Table 2, on the challenging real-world problem set, CausalM demonstrates clear advantages over other learning-based approaches. Specifically, it solves the largest number of instances in CORLAT (49), matching the performance of SCIP and outperforming other learning methods. Furthermore, CausalM delivers the highest solution quality among learning-based methods, as evidenced by the fewest explored nodes on both CORLAT and MIK. However, because the model introduces additional inference overhead, CausalM and other learning-based methods generally take longer to solve problems than heuristic approaches.

Table 3 evaluates the effectiveness of our data augmentation strategy under limited expert supervision. Specifically, CausalM-aug is trained on only 10% of the real data and 90% synthetic data, while all baselines are trained on 100% real data. Despite using far less supervision, CausalM-aug achieves competitive performance across all benchmarks, outperforming most heuristic and learning-based baselines on FCMCNF and GISP, and close to the best-performing methods on MAXSAT. Notably, CausalM-aug achieves 19 wins on FCMCNF and GISP, surpassing or matching fully supervised approaches. These results highlight the effectiveness of our augmentation strategy in addressing data scarcity, enabling the model to maintain low solve times and compact BnB trees even with limited oracle guidance. More broadly, this suggests that our method benefits not merely from more data, but from informative structural perturbations guided by optimality transitivity. The ability to generalize from synthetic data further supports the claim that CausalM captures the intrinsic structure of the BnB process, rather than overfitting to raw feature-label patterns.

## 4.3 Transfer Results

We evaluate the generalization capability of CausalM on larger MILP instances from the three benchmarks. As shown in Table 4, CausalM achieves the best overall performance, outperforming all learning-based and heuristic baselines on FCMCNF and MAXSAT, and ranking closely behind SCIP on GISP. Despite being trained only on smaller instances, CausalM achieves 44 wins on MAXSAT and 19 on FCMCNF, while fully supervised methods struggle to scale.

Specifically, on MAXSAT, CausalM slightly surpasses even the Oracle in both solve time and node count, demonstrating its ability to generalize beyond the training scale and perform efficient search in larger instances. On FCMCNF, it achieves the fastest solve time and highest win count, with only marginally more nodes than RankNet. Interestingly, RankNet explores fewer nodes but requires longer solving time. This highlights that simply reducing node count is not sufficient if the selected

Table 5: Comparison of CausalM-aug with baselines on transfer data.

| Methods | FCMCNF | | | MAXSAT | | | GISP | | |
|---|---|---|---|---|---|---|---|---|---|
| | Time(s) | Nodes | Wins | Time(s) | Nodes | Wins | Time(s) | Nodes | Wins |
| Oracle | 16.13 ± 1.72 | 74.59 ± 4.46 | - | 7.40 ± 1.49 | 160.33 ± 2.00 | - | 18.21 ± 1.53 | 1062.20 ± 1.81 | - |
| Estimate | 19.14 ± 1.93 | 122.37 ± 5.14 | 8 | 9.97 ± 1.55 | 247.01 ± 1.84 | 9 | 20.26 ± 1.68 | 1434.95 ± 1.84 | 1 |
| SCIP | 21.28 ± 1.84 | 178.45 ± 4.30 | 10 | 10.75 ± 1.34 | **170.92 ± 1.66** | 5 | **15.64 ± 1.48** | 1533.48 ± 1.85 | **37** |
| SVM | **18.30 ± 1.84** | 132.77 ± 4.73 | **13** | **8.80 ± 1.60** | 225.32 ± 2.06 | 11 | 18.53 ± 1.51 | 1221.79 ± 1.69 | 4 |
| RankNet | 19.76 ± 2.00 | **112.60 ± 5.50** | 3 | 9.50 ± 1.57 | 238.33 ± 2.11 | 3 | 18.89 ± 1.52 | 1215.05 ± 1.73 | 0 |
| L2C | 19.91 ± 1.90 | 120.17 ± 4.94 | 4 | 9.31 ± 1.70 | 233.10 ± 2.23 | 8 | 18.80 ± 1.52 | 1196.42 ± 1.75 | 0 |
| CausalM-aug | 19.59 ± 2.08 | 124.96 ± 5.19 | 12 | 8.97 ± 1.63 | 207.66 ± 2.09 | **14** | 17.17 ± 1.51 | **1182.28 ± 1.72** | 8 |

nodes contribute little to discovering feasible solutions or tightening bounds. In contrast, CausalM prioritizes nodes more likely to lie on the optimal path, improving both search depth and expansion utility. This results in faster bound convergence and more efficient overall search. On GISP, SCIP achieves the best performance, likely due to heuristic alignment with the problem structure. Nonetheless, CausalM remains competitive and consistently outperforms all other learning-based methods. These results confirm that structurally grounded causal modeling enables scalable and adaptive node selection, enabling robust performance across different problem scales and structures.

To further assess the generalization capability of our method under limited data, we evaluate CausalM-aug, trained with only 10% real data and 90% augmented data, on the transfer sets. Table 5 presents the results. Despite with only 10% supervision, CausalM-aug achieves strong performance across all benchmarks, demonstrating its ability to generalize to larger and more complex instances.

Notably, in GISP, CausalM-aug surpasses all other learning-based methods in both solve time and node count, and matching the fully supervised CausalM in wins (Table 4). This suggests that the augmentation strategy introduces sufficient structural variability to improve robustness on difficult instances. In MAXSAT, CausalM-aug achieves the highest win count among all baselines, even outperforming methods trained with full supervision such as L2C and SVM. This result highlights the quality of the synthetic data and confirms that our augmentation strategy can effectively mitigate data scarcity. Interestingly, while RankNet explores the fewest nodes in FCMCNF and GISP, its solve time remains higher than that of CausalM-aug. This reaffirms that simply reducing node count is not sufficient. CausalM-aug prioritizes nodes with stronger causal signals, which contribute more to discovering feasible solutions or tightening bounds, thereby accelerating convergence.

In summary, the results demonstrate that our data augmentation strategy not only reduces dependence on expert-labeled data, but also improves model generalization by exposing the learner to a wider range of decision structures. This leads to more effective node selection, better scaling to large instances, and stronger resilience to distribution shifts.

## 4.4 ABLATIONS

To evaluate the impact of our data augmentation strategy, we conduct ablation studies with three configurations: CausalM-10%, trained with only 10% of the original dataset; CausalM-aug, trained with 10% original data and 90% synthetic data generated by our augmentation method; and CausalM, trained with the full dataset. Table 6 presents the results on the test sets.

In all benchmarks, CausalM-aug outperforms CausalM-10% by a significant margin in both solve time and node count, demonstrating the effectiveness of data augmentation under limited supervision. Notably, CausalM-aug even surpasses the fully supervised CausalM in GISP, achieving the highest win count (28 vs. 20) and slightly better solve time, suggesting that synthetic data introduces beneficial diversity that enhances generalization to structurally complex or distribution-shifted instances. We also observe that CausalM-aug exhibits smaller standard deviations compared to CausalM-10%, indicating improved robustness and consistency across problem instances. These results confirm that the proposed augmentation strategy not only compensates for data scarcity, but also strengthens model adaptability and convergence stability.

Conversely, the degraded performance of CausalM-10% on MAXSAT (with the worst solve time and largest node count among the three) highlights that causal modeling alone is not sufficient when supervision is extremely limited. This further emphasizes the importance of either rich labeled data or effective augmentation to fully realize the benefits of causal reasoning in BnB node selection.

Table 6: Ablation study on test sets.

| Methods | FCMCNF | | | MAXSAT | | | GISP | | |
|---|---|---|---|---|---|---|---|---|---|
| | Time(s) | Nodes | Wins | Time(s) | Nodes | Wins | Time(s) | Nodes | Wins |
| CausalM-10 | 3.76 ± 1.51 | 26.76 ± 5.72 | 9 | 6.83 ± 1.73 | 175.38 ± 2.41 | 6 | 4.50 ± 1.24 | 286.83 ± 2.22 | 2 |
| CausalM-aug | 3.49 ± 1.52 | 21.23 ± 5.38 | **23** | 6.58 ± 1.79 | 132.94 ± 2.64 | 5 | **3.77 ± 1.24** | 206.67 ± 2.46 | **28** |
| CausalM | **3.43 ± 1.49** | **19.25 ± 5.07** | 18 | **5.28 ± 1.78** | **105.68 ± 2.50** | **39** | 3.82 ± 1.25 | **181.69 ± 2.77** | 20 |

Table 7: Ablation study on transfer sets.

| Methods | FCMCNF | | | MAXSAT | | | GISP | | |
|---|---|---|---|---|---|---|---|---|---|
| | Time(s) | Nodes | Wins | Time(s) | Nodes | Wins | Time(s) | Nodes | Wins |
| CausalM-10 | 21.22 ± 1.92 | 166.66 ± 4.40 | 10 | 8.83 ± 1.55 | 209.23 ± 2.10 | 5 | 20.62 ± 1.52 | 1323.70 ± 1.68 | 6 |
| CausalM-aug | 19.59 ± 2.08 | 124.96 ± 5.19 | 14 | 8.97 ± 1.63 | 207.66 ± 2.09 | 10 | **17.17 ± 1.51** | **1182.28 ± 1.72** | 15 |
| CausalM | **18.19 ± 1.84** | **114.16 ± 4.57** | **26** | **7.04 ± 1.49** | **155.93 ± 2.02** | **35** | 17.31 ± 1.51 | 1194.70 ± 1.72 | **29** |

Table 7 reports the ablation results on the transfer sets. Across all benchmarks, the augmented model (CausalM-aug) significantly outperforms CausalM-10%, achieving lower solving times and fewer explored nodes. In GISP, for instance, CausalM-aug reduces the node count from 1323.70 to 1182.28 and improves the solve time by over 3 seconds. These results confirm that the augmentation strategy not only improves in-distribution generalization, but also strengthens the model's ability to handle out-of-distribution, larger-scale problems.

Although the fully trained model (CausalM) achieves the best performance overall, CausalM-aug remains competitive, especially on GISP, where its solve time (17.17s) even slightly improves over CausalM (17.31s). This observation suggests that data augmentation may inject structural diversity into training, which helps the model generalize better to complex or underrepresented structures. Moreover, the win counts show that CausalM-aug closes much of the gap with the fully trained CausalM (e.g., 15 vs. 29 wins on GISP), while substantially outperforming CausalM-10% (6 wins). This demonstrates that effective augmentation can compensate for the lack of labeled data, yielding a scalable and generalizable node selection policy without requiring full supervision.

In summary, these results highlight that our augmentation strategy not only enhances performance but also promotes generalization to unseen problems, which is key for practical deployment where labeled data is limited.

## 5 Conclusion

In this paper, we have presented causal modeling for node selection in BnB for solving NP-hard MILPs. Our method builds on the observation that optimal solutions propagate along the BnB tree, and explicitly models this causal structure via contrastive learning. Extensive experiments across multiple benchmarks and real-world problems demonstrate that our approach significantly reduces both solving time and BnB tree size, outperforming heuristic and learning-based baselines.

Beyond empirical gains, our framework provides a structurally grounded perspective on node selection. Rather than relying on heuristic indicators or feature-level correlations, the model is guided by a search-tree property that reflects how optimality evolves within BnB. This perspective opens the door to more principled learning objectives for combinatorial solvers and suggests future directions in designing structure-aware policies that generalize across instance types and solver configurations.

A promising future direction is to explore stronger expert strategies or self-improving learners beyond imitation of a fixed oracle. In particular, mixed-integer linear programming (MILP) problems can have multiple optimal solutions and optimal paths, and understanding how to efficiently leverage these multiple optima could be an important avenue for future research. Distilling structural insights from diverse optimal solutions and adapting expert knowledge dynamically into the learning process may further enhance efficiency and robustness.

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

# A APPENDIX

## A.1 RELATED WORK ON CONTRASTIVE LEARNING

Self-supervised learning has emerged as a powerful paradigm for representation learning, where models are trained on automatically constructed pretext tasks without requiring manual labels (Jing & Tian 2020). Among various self-supervised learning approaches, contrastive learning has achieved remarkable success in visual understanding (He et al. 2020, Chen et al. 2020, Yang et al. 2023) and graph representation learning (You et al. 2020, Tong et al. 2021, Wang et al. 2022). By contrasting positive (semantically similar) and negative (dissimilar) pairs, it induces representations where similar items are pulled together and dissimilar ones pushed apart.

Recent works have explored contrastive learning in optimization. Mulamba et al. (2020) introduced a contrastive loss in decision-focused learning for combinatorial problems with uncertain inputs. The optimal and non-optimal solutions were treated as positives and negatives, respectively. Duan et al. (2022) employed contrastive pretraining to Boolean satisfiability, inspired by its success in computer vision. Huang et al. (2023) integrated contrastive learning with large neighborhood search, identifying near-optimal solutions as positive to guide neighborhood prediction. Despite these advances, contrastive learning remains underexplored in BnB node selection. Its pairwise training algins naturally with the goal of distinguishing optimal from non-optimal nodes, enabling quality-aware representations without explicit supervision.

## A.2 TECHNICAL DETAILS OF THE CAUSALITY MODELING

### A.2.1 DATA COLLECTION

We collect training data from MILP instances, each of which has one or more known optimal solutions. For each branching step $t$ of each instance, we construct data points using a diving oracle. Each data point is a triplet $(v_t^+, v_t^{-,k}, (v_t^{-,k})^{\text{anc}})$, $k = 1, 2, \cdots, |\mathcal{V}_t| - 1$. Here, $\mathcal{V}_t$ denotes the candidate node set at $t$; $v_t^+ \in \mathcal{V}_t$ is a positive candidate node that contains an optimal solution; $v_t^{-,k} \in \mathcal{V}_t \setminus \{v_t^+\}$ is the $k$th negative node; $(v_t^{-,k})^{\text{anc}}$ is the lowest common ancestor of $v_t^+$ and $v_t^{-,k}$ in the BnB tree. The lowest common ancestor is not necessarily their immediate parent, although it may coincide with one in special cases. We use the lowest common ancestor instead of the root as the anchor, as it provides a more localized context that better preserves the branching path relevant to the contrast. Figure 1 illustrates the data construction.

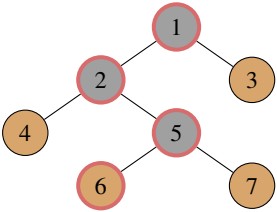

Figure 1: Illustration of data construction. Nodes with red borders contain an optimal solution. Nodes in yellow are candidates, in which node 6 is positive and nodes 3, 4, 7 are negative. Nodes 6 and 4 share the lowest common ancestor node 2; 6 and 7 share 5; 6 and 3 share 1. Three triplets are constructed: (node 6, node 4, node 2), (node 6, node 7, node 5), and (node 6, node 3, node 1).

### A.2.2 DATA AUGMENTATION

Given the positive node $v_t^+$ and its ancestor $(v_t^{-,k})^{\text{anc}}$, we represent the branching path between them as a sequence of decisions

$$v_t^+ = \psi\big((v_t^{-,k})^{\text{anc}}, d_1, d_2, \ldots, d_L\big), \tag{12}$$

where $\psi$ maps the sequence of decisions $d_1, d_2, \ldots, d_L$ from $(v_t^{-,k})^{\text{anc}}$ to $v_t^+$. We modify this path by perturbing each decision with random noise, including flipping the inequality direction or

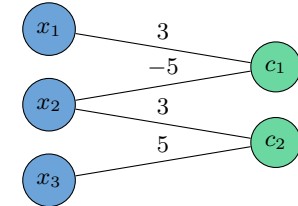

Minimize: $-2x_1 - x_2 + 3x_3$
s.t.
$c_1 : \ 3x_1 - 5x_2 \le 0$
$c_2 : \ 3x_2 + 5x_3 \le 15$
$x_1, x_2, x_3 \in Z$

Figure 2: Illustration of bipartite graph representation of BnB node. Variables $x_1, x_2, x_3 \in V$ and constraints $c_1, c_2 \in C$ are represented as nodes, with edge weights $3, -5, 3, 5 \in E$ indicating coefficients in the constraint matrix.

perturbing the branching cutoff value. An augmented node is then obtained by

$$v_t^{+'} = \psi\big((v_t^{-,k})^{\text{anc}}, \Delta(d_1), \Delta(d_2), \dots, \Delta(d_L)\big). \tag{13}$$

where $\Delta(\cdot)$ refers to the perturbation. We apply rejection sampling to discard any modified path that violates $\mathbf{x}^* \in \mathcal{F}_{v_t^{+'}}$. The same procedure applies to each negative node $v_t^{-,k}$, generating an augmented negative $v_t^{-,k'}$ such that $\mathbf{x}^* \notin \mathcal{F}_{v_t^{-,k'}}$. This yields augmented triplets $\big(v_t^{+'}, v_t^{-,k'}, (v_t^{-,k})^{\text{anc}}\big)$ for all $k = 1, 2, \dots, |\mathcal{V}_t| - 1$. The augmentation increases training diversity without additional oracle queries, improving model robustness and reducing computational overhead.

### A.2.3 MODEL ARCHITECTURE

Following prior work (Gasse et al. 2019, Labassi et al. 2022, Scavuzzo et al. 2022), we represent each node as a bipartite graph $G = (E, V, C)$. Here, $E \in \mathbb{R}^{|V| \times |C|}$ encodes the connectivity between variables and constraints; an edge $(p, q)$ exists if the $p$th variable appears in the $q$th constraint, and the edge value is the coefficient $A_{q,p}$ from the constraint matrix $\mathbf{A}$ in equation 1. $V \in \mathbb{R}^{|V| \times d_V}$ contains variable node attributes, including type (binary, integer, continuous), objective coefficients, and bounds. $C \in \mathbb{R}^{|C| \times d_C}$ contains constraint node attributes, including constraint type ($\le, =, \ge$) and right-hand side value. Figure 2 illustrates the bipartite graph.

With the bipartite graph representation $G = (E, V, C)$, we encode each node using the encoder $\phi_\theta$. In $\phi_\theta$, the raw graph representation is firstly mapped into a latent space using three multi-layer perceptrons (MLPs):

$$\mathbf{E} = \text{MLP}_E(E), \quad \mathbf{V} = \text{MLP}_V(V), \quad \mathbf{C} = \text{MLP}_C(C), \tag{14}$$

where $\mathbf{E} \in \mathbb{R}^{|C| \times |V|}$, $\mathbf{V} \in \mathbb{R}^{|V| \times d}$, and $\mathbf{C} \in \mathbb{R}^{|C| \times d}$ are the edge, variable, and constraint embeddings, respectively.

Then, a GNN is applied, in which each layer alternates between updating constraint and variable embeddings. The update rule is defined as

$$\mathbf{C}' = \sigma\left(\mathbf{W_1}\mathbf{C} + \mathbf{W_2}\sum_{p \in \mathcal{N}(q)} \mathbf{E}_{q,p} \cdot \mathbf{V}_p\right), \quad \mathbf{V}' = \sigma\left(\mathbf{W_1}\mathbf{V} + \mathbf{W_2}\sum_{q \in \mathcal{N}(p)} \mathbf{E}_{q,p} \cdot \mathbf{C}_q\right), \tag{15}$$

where $\mathbf{W_1}, \mathbf{W_2} \in \mathbb{R}^{d \times d}$ are shared learnable weights, $\sigma$ is a nonlinear activation, $\mathcal{N}(q)$ denotes the set of variables connected to the $q$th constraint, and $\mathcal{N}(p)$ denotes the set of constraints connected to the $p$th variable.

Finally, average pooling is applied over the variable embeddings

$$\mathbf{v}_{\text{avg}} = \frac{1}{|\mathcal{V}|}\sum_{p=1}^{|\mathcal{V}|} \mathbf{V}'_p, \tag{16}$$

where $\mathbf{v}_{\text{avg}} \in \mathbb{R}^d$ is the final embedding of the node. As constraint information has been integrated into $\mathbf{V}'$, no additional aggregation is needed.

A.3    THEORETICAL ANALYSIS OF THE CAUSALITY MODELING

This section provides the complete theoretical analysis of why the contrastive loss aligns with the causal signal of node optimality.

**Definition 2** (Causal estimand). *For a node $v$ and its ancestor $v^{anc}$, let $y \in \{0, 1\}$ denote whether the subtree rooted at $v$ contains an optimal solution under the intervention of expanding $v$. The causal target of node selection is to rank the candidate successors (children) of $v^{anc}$ by $P(y = 1 \mid v^{anc}, v)$.*

**Assumption 1** (Conditional exchangeability). *Positives $v^+$ are sampled from $p^+(v \mid v^{anc}) = p(v \mid y = 1, v^{anc})$ and negatives $v^-$ from $p^-(v \mid v^{anc}) = p(v \mid y = 0, v^{anc})$. We assume conditional exchangeability: given $v^{anc}$, the sampling of $(v^+, v^-)$ does not depend on latent confounders that also affect $y$.*

**Assumption 2** (Positivity and label-conditional sampling). *For any $v$ in the candidate set of a fixed $v^{anc}$, $p^+(v \mid v^{anc}) > 0$ iff $p^-(v \mid v^{anc}) > 0$ (positivity). The triplet construction is label-conditional. That is, the sampling distribution of $(v^+, v^-)$ given $(y, v^{anc})$ is proportional to the observational class-conditionals $p(v \mid y, v^{anc})$ up to a $y$-dependent factor that does not depend on $v$.*

Define the score $Q(v^{\mathrm{anc}}, v) \triangleq \frac{H(v^{\mathrm{anc}}, v)}{\tau}$, and the two-way InfoNCE loss

$$\mathcal{L} = \mathbb{E}_{v^{\mathrm{anc}}, v^+, v^-} \, \log\big(1 + \exp(Q(v^{\mathrm{anc}}, v^-) - Q(v^{\mathrm{anc}}, v^+))\big). \tag{17}$$

**Lemma 1** (Bayes-optimal solution). *The minimizer of $\mathcal{L}$ satisfies*

$$Q^\star(v) = \log \frac{p^+(v \mid v^{anc})}{p^-(v \mid v^{anc})} + c(v^{anc}),$$

*where $c$ is an ancestor-dependent constant.*

*Sketch.* The pairwise logistic loss is minimized when the score difference matches the log-odds of positive vs. negative samples, as in InfoNCE. This yields the stated solution up to an additive constant (Joachims 2002, Oord et al. 2018). The invariance to class priors follows from the fact that the optimal logit depends only on the class-conditional density ratio. □

**Proposition 2** (Ranking consistency). *Under Assumptions 1–2, $Q^\star(v)$ is a monotone transform of $\log \frac{P(y=1|v^{anc},v)}{P(y=0|v^{anc},v)}$. Hence, ranking by $Q^\star$ coincides with ranking by the causal probability $P(y = 1 \mid v^{anc}, v)$.*

**Theorem 2** (Fisher consistency). *If the function class contains $Q^\star$, any sequence $\{Q_t\}$ with $\mathcal{L}(Q_t) \to \inf_Q \mathcal{L}(Q)$ yields rankings converging (in probability) to those induced by $P(y = 1 \mid v^{anc}, v)$ for almost every $v^{anc}$.*

*Sketch.* Follows from Lemma 1 and standard results on classification-calibrated surrogate losses (Bartlett et al. 2006). □

**Mutual-information view.**    Let $r \in \{+, -\}$ denote whether $v$ is sampled from $p^+$ or $p^-$. Negatives are assumed i.i.d. from the conditional marginal $p(v \mid v^{\mathrm{anc}})$. For the two-way InfoNCE loss, the optimal value satisfies

$$I(r; V \mid V^{\mathrm{anc}}) \geq \log 2 - \mathcal{L},$$

with equality at the Bayes optimum (Oord et al. 2018). Thus minimizing $\mathcal{L}$ maximizes a lower bound on the conditional mutual information $I(r; V \mid V^{\mathrm{anc}})$, thereby increasing the separability of positive vs. negative children.

**Role of temperature.**    The temperature $\tau$ rescales $Q$ but does not change the Bayes-optimal ranking, as any positive scaling preserves order. It acts as a margin-softness parameter that trades off gradient magnitude and numerical stability.

**Generalization.** Let $\mathcal{G} = \{ g((v^{\text{anc}}, v^+, v^-)) = Q(v^{\text{anc}}, v^-) - Q(v^{\text{anc}}, v^+) : Q \in \mathcal{Q} \}$ be the pairwise-difference class induced by $\mathcal{Q}$. Define its empirical Rademacher complexity over $\tilde{n}$ triplets $\{z_i\}_{i=1}^{\tilde{n}}$ by

$$\mathfrak{R}_{\tilde{n}}(\mathcal{G}) = \mathbb{E}_\sigma \Big[ \sup_{g \in \mathcal{G}} \tfrac{1}{\tilde{n}} \sum_{i=1}^{\tilde{n}} \sigma_i \, g(z_i) \Big].$$

By standard contraction arguments for Lipschitz losses, the empirical minimizer of equation 17 generalizes provided $\mathfrak{R}_{\tilde{n}}(\mathcal{G})$ is controlled (e.g., via norm constraints on $Q$) (Bartlett & Mendelson 2002).

**Causal perspective and limitations.** The analysis shows that the InfoNCE loss is statistically well-founded and, under Assumptions 1–2, aligns with the causal estimand of node optimality. However, identifiability conditions (e.g., full back-door admissibility) may not be strictly satisfied in practice, so the causal view should be regarded as an idealized principle. In real applications, the objective can be interpreted as a robust surrogate that reduces but does not fully eliminate confounding bias. This principled, yet approximate alignment, is what differentiates our approach from correlation-based models.

### A.4 DETAILED EXPERIMENTAL SETUP

#### A.4.1 BENCHMARKS

We evaluate the proposed causality modeling approach (CausalM) on three widely-used NP-hard MILP benchmarks, following the experimental setup of Labassi et al. (2022). Each benchmark contains training, validation, test, and transfer sets. The training, validation, and test sets share the same size distribution, while transfer sets consist of larger instances for investigating generalizability. The benchmarks are:

- Fixed charge multicommodity network flow (FCMCNF) (Hewitt et al. 2010): generated using Chmiela et al. (2021)'s code. We use instances with 20 nodes and 30 commodities for training, validation, and test, 30 nodes and 45 commodities for transfer.
- Maximum satisfiability (MAXSAT) (Ansótegui & Gabàs 2017): generated via the method in Béjar et al. (2009). Numbers of nodes in training, validation, and test instances are sampled from $[60, 70]$, and transfer from $[80, 100]$.
- Generalized independent set (GISP) (Colombi et al. 2017): generated using Chmiela et al. (2021)'s code. Numbers of nodes in training, validation, and test instances are sampled from $[60, 70]$, and transfer from $[70, 80]$.

All above problems rely on an underlying graph structure, generated by Erdős–Rényi random graphs (Erdos et al. 1960). The edge probability is $0.33$ for FCMCNF, and $0.6$ for MAXSAT and GISP.

In addition, we also evaluate on two real-world datasets:

- CORLAT (Gomes et al. 2008): a real-world dataset for constructing a wildlife corridor for grizzly bears in the Northern Rockies region.
- MIK (Atamtürk 2003): a set of MILP problems with knapsack constraints.

Table 8 summarizes the statistics of the MILP problems used in our experiments, including the number of instances and average problem sizes in terms of binary, integer, and continuous variables, as well as constraints.

#### A.4.2 BASELINES

We compare against the following baselines:

- Oracle: an expert diving policy constructed by first solving the MILP to obtain the optimal solution and then, during the BnB process, always selecting the active node whose feasible region contains this optimal solution.
- Estimate: the widely used heuristic based on best estimate search.

Table 8: Statistics of MILP problem instances.

| Metric | FCMCNF | | MAXSAT | | GISP | | CORLAT | MIK |
| | Test | Transfer | Test | Transfer | Test | Transfer | Test | Test |
|---|---|---|---|---|---|---|---|---|
| Avg Binary | 64.48 | 115.20 | 1426.00 | 1722.92 | 64.80 | 90.80 | 100.00 | 75.00 |
| Avg Integer | 0.00 | 0.00 | 0.00 | 0.00 | 0.00 | 0.00 | 0.00 | 300.00 |
| Avg Continuous | 1418.56 | 3456.00 | 0.00 | 0.00 | 623.66 | 1229.84 | 366.00 | 11.67 |
| Avg Constraints | 394.48 | 715.20 | 1361.16 | 1647.92 | 1247.66 | 2458.70 | 484.92 | 311.67 |
| Num Instances | 50 | 50 | 50 | 50 | 50 | 50 | 50 | 18 |

- SCIP (Gamrath et al. 2020): SCIP's default node selection, which augments Estimate with a diving heuristic.
- SVM (He et al. 2014): an imitation learning method based on the support vector machine.
- RankNet (Song et al. 2018): a MLP trained with pairwise ranking loss.
- L2C (Labassi et al. 2022): the state-of-the-art node selection method using a GNN.

### A.4.3 METRICS

We report the following evaluation metrics (Labassi et al. 2022, Zhang et al. 2024):

- Time: total solving time in seconds.
- Nodes: number of nodes in the final BnB tree.
- Wins: A "win" is counted for a method on a given instance if it achieves the smallest solving time among all compared methods.
- Solved: number of instances solved with timelime of 3600 seconds.

Time and Nodes are averaged using the shifted geometric mean with a shift of 1, following the common MILP evaluation practice. For unsolved instances, we take SCIP's final node count and time to include them in the averaged metrics.

### A.4.4 IMPLEMENTATION

We generate 1000 instances per benchmark for training. This results in 16285, 41299, and 42511 training triplets for FCMCNF, MAXSAT, and GISP, respectively. Since learning-based methods are trained to imitate Oracle's decisions, we also generate 100 instances per benchmark for validating the imitation accuracy. This results in 3019, 4868, and 5236 validation triplets for FCMCNF, MAXSAT, and GISP, respectively. For each benchmark, 50 in-distribution test instances and 50 out-of-distribution transfer instances are generated to evaluate model performance and generalization.

For the real-world datasets, we used 1,000 CORLAT instances for training, 100 for validation, and 50 for testing. For the MIK dataset, we split a total of 90 instances into 72 for training and 18 for validation and testing. In terms of sample size, the CORLAT training set contains 143,593 samples, while the MIK training set contains 6,578 samples.

In CausalM, each of the three MLP blocks consists of a LayerNorm, a linear layer, and a ReLU activation; the GCN is three-layer and the activation $\sigma$ is also ReLU. The temperature $\tau$ in equation 11 is set to 0.07, following the default setting in He et al. (2020). The model is trained using Adam (Kingma & Ba 2014) with a batch size of 64 and a learning rate of $5 \times 10^{-3}$. For the data augmentation strategy in Section A.2.2, we modify each branching decision in equation 12 by replacing it with an alternative candidate branch variable at that node. For positive samples, we ensure that the chosen branching direction (e.g., $x > 1$ or $x < 0$) leads to a subproblem that contains the optimal solution. For negative samples, the branching direction is selected randomly. We test two variants of our model: (i) CausalM, trained on the full dataset generated, and (ii) CausalM-aug, trained with only 10% of the original data and 90% synthetic data generated via the data augmentation strategy to access the effectiveness of the augmentation. SVM, RankNet, and L2C are faithfully reproduced using the code from Labassi et al. (2022), without additional tuning or modification. All methods are implemented in SCIP 7.0.3 with default parameters. The experiments are conducted on a workstation equipped with an Intel(R) Xeon(R) Gold 6348 CPU @ 2.60GHz and a single NVIDIA GeForce

Table 9: Valid accuracy of learning-based approaches in imitating the diving oracle.

|  | FCMCNF | MAXSAT | GISP | CORLAT | MIK |
|---|---|---|---|---|---|
| SVM | 0.926 | 0.934 | 0.965 | 0.973 | 0.862 |
| RankNet | 0.976 | 0.959 | 0.978 | 0.994 | 0.954 |
| L2C | 0.945 | 0.960 | 0.978 | 0.999 | 0.856 |
| CausalM | 0.951 | 0.987 | 0.971 | 0.996 | 0.907 |

RTX 3090 GPU. Model training is performed on the GPU, while inference is executed on a single CPU thread with a fixed random seed to ensure reproducibility.

### A.4.5 INFERENCE

We use the learned similarity function $H$ to define a node comparison strategy for BnB. Specifically, our model replaces SCIP's (Gamrath et al. 2020) default `NODECOMP` function, which compares two candidate nodes. Since ranking all candidates at each selection step is computationally expensive, SCIP maintains a priority queue of candidates. When new nodes are created, they are inserted into the queue based on `NODECOMP`'s pair-wise comparison outcome. The `NODESELECT` function then selects the next node to explore according to the queue. We redefine `NODECOMP` using our model as

$$\text{NODECOMP}(v_a, v_b, v^{\text{anc}}) = \begin{cases} -1 & \text{if } H(v_a, v^{\text{anc}}) > H(v_b, v^{\text{anc}}); \\ 1 & \text{if } H(v_b, v^{\text{anc}}) > H(v_a, v^{\text{anc}}); \\ 0 & \text{otherwise}, \end{cases} \tag{18}$$

where $v_a$ and $v_b$ are candidate nodes and $v^{\text{anc}}$ is their lowest common ancestor. The returned value follows SCIP's convention, i.e., $-1$ prefers $v_a$, $1$ prefers $v_b$, and $0$ indicates no preference. This allows integrating causality-driven node selection into existing solvers with minimal overhead.

### A.5 ADDITIONAL EXPERIMENTAL RESULTS

We report the imitation accuracy on validation data in Table 9, i.e., how often each method selects the same node as the diving Oracle on held-out instances. While RankNet and L2C achieve high imitation accuracy, their search performance, as reported in Table 10, are inferior. That is, the high imitation accuracy suggests generalization in mimicking Oracle decisions, but the correlation-based modeling fails to translate this imitation into effective search. This supports our analysis of the mismatch in Section 3.1. In comparison, CausalM achieves both high accuracy and strong performance in terms of effectiveness and efficiency. These results confirm that modeling causal signals, rather than fitting feature-label correlations, leads to more effective and generalizable models.

Table 10: Comparison of CausalM with baselines on standard test datasets for different temperature values ($\tau$).

| $\tau$ | FCMCNF | | MAXSAT | | GISP | |
|---|---|---|---|---|---|---|
| | Acc | Nodes | Acc | Nodes | Acc | Nodes |
| 0.03 | 0.951 | $21.51 \pm 5.65$ | 0.985 | $110.82 \pm 2.52$ | 0.978 | $176.68 \pm 2.72$ |
| 0.05 | 0.950 | $18.99 \pm 5.54$ | 0.983 | $105.77 \pm 2.41$ | 0.985 | $226.31 \pm 2.54$ |
| 0.07 | 0.951 | $19.25 \pm 5.07$ | 0.987 | $105.68 \pm 2.50$ | 0.971 | $181.69 \pm 2.77$ |
| 0.10 | 0.943 | $20.71 \pm 5.49$ | 0.984 | $106.86 \pm 2.51$ | 0.965 | $202.98 \pm 2.74$ |

We further report the mean optimality gap of each method, which measures the relative difference between the best-found feasible solution and the optimal one. As a standard indicator of solution quality and convergence, a smaller gap implies that the method converges faster and achieves better performance within the time limit. As shown in Figure 3, CausalM consistently achieves smaller optimality gaps across benchmarks, highlighting its superior convergence efficiency and solution quality compared to all baselines.

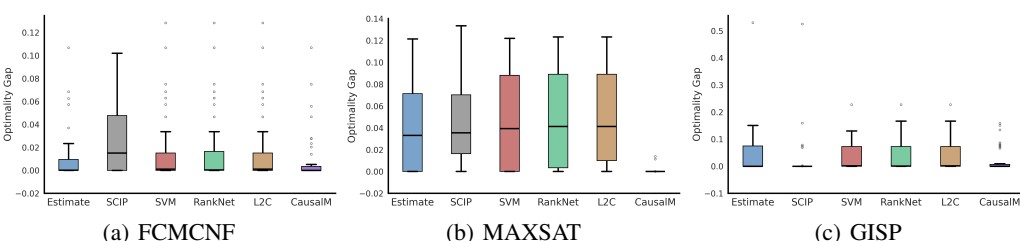

Figure 3: Optimality gap across benchmark problems over test sets.

# B    LLM USAGE

Large language models (LLMs) were used in the preparation of this manuscript for polishing and writing assistance, including grammar correction and improving readability. All research ideas, study design, analyses, results, and conclusions are the sole work of the authors. The LLM did not contribute to the scientific content or interpretation.

