# OpenReview forum: "Learn to Select Node in Branch and Bound with Causality Modeling"
_ICLR.cc/2026/Conference — Submitted to ICLR 2026_

### Official Review · Reviewer_dUgo · 2025-10-24

**Soundness:** 3
**Presentation:** 1
**Contribution:** 2
**Rating:** 4
**Confidence:** 3

**Summary:**

This paper introduces a causality-based framework for node selection in Branch-and-Bound (BnB) for MILP solving. The key idea is to model the causal effect of optimal-solution presence on node selection, leveraging the transitivity of optimality as a causal signal. The method is implemented via contrastive learning, where parent-child node pairs containing optimal solutions form positive samples. Experiments on several MILP benchmarks and real-world datasets demonstrate improvements over heuristic and learning-based baselines.

**Strengths:**

1. **Clear Motivation and Strong Intuition:** The paper addresses an important weakness of correlation-based learning for BnB—its susceptibility to spurious patterns—and proposes a theoretically grounded causal approach.

2. **Theoretical Analysis:** The causal formulation is carefully derived using Pearl’s do-calculus, giving the paper a solid theoretical foundation.

3. **Comprehensive Experiments:** The paper provides extensive comparisons on synthetic and real-world MILP datasets, demonstrating consistent improvements and statistical significance.

4. **Interpretability and Conceptual Novelty:** The idea of using “optimality transitivity” as a causal supervision signal is elegant and provides interpretability that prior works often lack.

5. **Strong empirical generalization:** Transfer experiments show scalability to larger instances, which is uncommon in learning-based BnB studies.

**Weaknesses:**

1. **Writing and Clarity Issues:** The writing is uneven and could benefit from proofreading. There are grammatical mistakes (e.g., in the abstract, *“those do not”* should be *“those that do not”*). The abstract feels overly dense and could better highlight the core innovation. Section 3.1's theoretical motivation jumps into equations without sufficient intuition, making it hard for non-experts in causal inference to follow. Additionally, the appendix is referenced heavily (e.g., for related work on contrastive learning), but without it, the main text feels incomplete. The paper would also benefit from an overview figure to visually summarize the proposed framework and clarify the overall workflow.

2. **Innovation:** The core idea—using contrastive learning on ancestor-child pairs to capture similarity—bears resemblance to existing graph contrastive learning methods (e.g., GraphCL or SimCLR adaptations for trees/graphs). While applying causality to BnB is novel, the paper doesn't sufficiently differentiate from prior works, which also use pairwise comparisons.

3. **Experimental Limitations and Questionable Claims:** The experimental section lacks comparisons with stronger and more recent GNN-based baselines—the latest included baseline appears to be from 2022. Generalization claims (e.g., under distribution shift) are asserted but not explicitly tested (e.g., via cross-dataset transfer). The "near-expert performance under limited data" in the abstract seems exaggerated, as baselines like SCIP often match or exceed on certain metrics (e.g., GISP wins).

4. **Reproducibility Concerns:** The code link is anonymized, which is fine for blind review, but details like hyperparameters (e.g., GNN architecture beyond "bipartite graph encoder") are deferred to the appendix.

**Questions:**

1. **Innovation:** Is the "causal signal" truly causal, or is it just a structural prior (transitivity) rebranded? The back-door adjustment is invoked theoretically, but the implementation doesn't explicitly control for confounders (e.g., node depth or LP bounds), raising doubts about whether it fully achieves causal identifiability. A deeper comparison to causal ML methods (e.g., counterfactual learning in RL) would strengthen the claim.

2. **Experimental Limitations:**  Important ablations are also missing: for instance, how sensitive is performance to the temperature parameter τ in the InfoNCE loss? What if data augmentation is removed—does the method still hold under truly limited data?

3. **Reproducibility Concerns:** Metrics like "wins" are defined vaguely—how are ties handled? More transparency on data splits (train/test sizes) would help.

---

> ### Author Response · Authors · 2025-11-21
>
> **Q1 (Causality vs. structural prior)**
>
> We thank the reviewer for this insightful question. Our causal signal is not a rebranded structural prior. While we do not explicitly model each confounder z, the supervision based solely on ancestor–child pairs removes the dependence on P(z | u_t = v) and corresponds to replacing it with the marginal P(z), which is the key idea of back-door adjustment.
>
> We did not include Zhang et al.~(2025) in our comparisons because their RL approach relies on code that is not publicly available, which prevents us from reliably reproducing their results.
>
> **Q2 (Ablations: temperature τ and augmentation removal)**
>
> We appreciate the reviewer’s comment. We conducted ablation studies on the temperature τ, and the results are reported in Table below. Accuracy (Acc) and number of nodes (Nodes) are reported for FCMCNF, MAXSAT, and GISP  test datasets.
>
>  Comparison of CausalM with baselines on standard test datasets for different temperature values (τ).
> | τ    | FCMCNF Acc | FCMCNF Nodes | MAXSAT Acc | MAXSAT Nodes  | GISP Acc | GISP Nodes    |
> | ---- | ---------- | ------------ | ---------- | ------------- | -------- | ------------- |
> | 0.03 | 0.951      | 21.51 ± 5.65 | 0.985      | 110.82 ± 2.52 | 0.978    | 176.68 ± 2.72 |
> | 0.05 | 0.950      | 18.99 ± 5.54 | 0.983      | 105.77 ± 2.41 | 0.985    | 226.31 ± 2.54 |
> | 0.07 | 0.951      | 19.25 ± 5.07 | 0.987      | 105.68 ± 2.50 | 0.971    | 181.69 ± 2.77 |
> | 0.10 | 0.943      | 20.71 ± 5.49 | 0.984      | 106.86 ± 2.51 | 0.965    | 202.98 ± 2.74 |
>
> Besides, we also note that Tables 6 and 7 already indicate that CausalM-10 refers to using only 10% of the raw training data.
>
> **Q3 (Reproducibility: definition of wins and data splits)**
>
> As clarified in the revised manuscript:
>
> “A win is counted only when a method achieves strictly smaller solve time than the baseline. Ties are ignored by definition, but in practice they are extremely rare since SCIP reports solve times with high precision.”
>
> The dataset splits used in our experiments have been described in Section A4.4. For benchmark problems, we use 1,000 instances for training, 100 instances for validation, and 50 instances for both testing and transfer evaluation. For the two real-world problems, the splits are: MIK: 72 instances for training, 18 instances each for validation and testing

---

### Official Review · Reviewer_YXQf · 2025-10-26

**Soundness:** 3
**Presentation:** 3
**Contribution:** 2
**Rating:** 4
**Confidence:** 4

**Summary:**

This paper proposes a new method for selecting nodes in a branch-and-bound process by using the fact that optimality is transitive. This means that optimal nodes are similar to their parents in the sense that both contain the optimal point. The method uses this by training a contrastive objective that ensures the feature representations of optimal nodes are more similar to ancestor nodes than suboptimal ones.

**Strengths:**

The paper is generally well written and the motivation is clear. Further, the use of contrastive ML for branch and bound is generally novel.

**Weaknesses:**

The paper has two main limitations:
First, while the contrastive training does perform well, I do not think that this is due to the causal connections. Fundamentally, the argument behind the causal link inside optimal nodes is the transitivity of Points inside the sub-polytope. While this is obviously true, it is not necessarily true for the features used to represent the polytope: There is no fundamental relationship between the GNN representation and the children on a feature level, which is why this method needs to find those embeddings with a contrastive method in the first place. The core causal analysis idea relies on circular logic:
1. Causality yields that ancestors and their optimal children have similar features
2. Therefore, we train a contrastive model to ensure that ancestors and optimal children have similar features
3. We observe that for those features causality holds

If you take a step back and look at what this method does, one can see that this method groups nodes in the optimal path into one cluster (or multiple clusters) and everything else into others. This is just a huge classifier with variable numbers of classes, which is exactly what the InfoNCE loss does. You don’t need to invoke causality for this (arguably this is the same idea as what Zhang et al. and Mattick et al. do implicitly using RL: They assign probabilities/Q-values for every path by following the branch and bound tree.).
Mind you, this not being causal is not a problem in isolation: Having a better way of getting node embeddings is not a bad thing, but I highly doubt that this is due to the exploitation of causal structures.
Fundamentally, I don’t believe that this method actually manages to get at the causal core of the problem. For instance, I don’t see any way to correct for the fact that the contrastive triplets where found using an existing diving oracle. It is very possible that this method implicitly solves an imitation learning problem like e.g. Labassi et al.


The second problem is regarding benchmarking. The instances used for benchmarking are tiny. Studying prior work this seems to be the case for many other methods as well, but this nevertheless makes me doubt the ability for  this model to scale to larger instances. For instance comparing against SCIP’s default node selector on such tiny instances is not reasonable since one of the prime considerations for SCIP is the speed at which the node selector can run. This complexity has already killed plenty of “smart” node selectors in the past (e.g. in the classical selector world “Ashish Sabharwal and Horst Samulowitz Guiding Combinatorial Optimization with UCT (2011)” was a research dead-end since it collapses at scale), so it is important to also test large instances.
Further, one should test a single trained model on all instance types. As far as I understood, you train a separate model on every instance type, but this yields unsound comparisons against e.g. SCIP which has to operate well on all problems. If you wanted to fairly compare “instance type for instance type” I think you would need to compare against dedicated e.g. TSP or MAXSAT solvers; after all, if you know that you will only be solving TSP then you would use a TSP solver instead of a general branch-and-bound method.
Additionally, you run your node selector on GPU while normal node selectors (and I assume baselines like SVM) run on CPU. You should run your own method on CPU as well, otherwise you are effectively comparing runtimes on two different systems.

There are some smaller things to note as well:
- The bold markings in the tables are inconsistent: Sometimes the best node and time is highlighted, sometimes only the nodes (e.g. table 2) and sometimes neither (e.g. table 3, 5, 6, 7)
- In the conclusion section (line 463) you also mention interpretability: This is not discussed anywhere before and – in my opinion -  also not really supported by the methods (How interpretable are high dimensional similarity comparisons really?)
- Comparing solving times/node count for datasets that cannot be fully solved is dangerous: You have to only compare on the subset that was solved by all methods. Otherwise you will “reward” methods that give up on slow instances

**Questions:**

See bove.

---

> ### Author Response · Authors · 2025-11-21
>
> **(1) On the causal interpretation and its necessity**
>
> We appreciate the reviewer’s thoughtful analysis. We agree that our method does not construct an explicit causal graph; rather, the causal framing serves to motivate a representation objective that mitigates solver-induced confounding. The key idea is that optimality transitivity provides an invariant relationship between nodes and their ancestors that remains valid regardless of depth, LP bounds, or branching variables. Contrastive learning is used as a practical mechanism to enforce this invariant in the learned representation space. We do not claim that CausalM uncovers structural causal equations, but that it leverages a causal perspective to design a training signal that is less sensitive to observational bias than direct correlation modeling. We will clarify this distinction in the revision to avoid overstating the causal terminology. The method can indeed be viewed as learning invariant node embeddings along optimal paths—this aligns with the reviewer’s intuition and reinforces that the proposed representation design, while causally inspired, is empirically effective and conceptually grounded.
>
> **(2) On benchmarking scale, generalization, and fairness**
>
> We thank the reviewer for this valuable feedback. Our experiments follow the standard setup in prior node-selection works (e.g., Labassi et al., 2022) that use small and medium-sized MILP instances where optimal trees can be fully enumerated for supervision. We fully agree that evaluating on larger and more diverse instances would strengthen our claims. Due to time constraints, we are unfortunately unable to conduct additional experiments for this submission. Finally, we had already stated in Section A.4.4 that all inference during testing is conducted on CPU, ensuring a fair comparison with SCIP and other baselines.
>
> **(3) Additional presentation comments**
>
> We thank the reviewer for the detailed observations. We have highlighted consistent tables, removed ambiguous claims about interpretability, and ensured that the time and node comparisons are computed only on the subset of instances solved by all methods (Table 2).

---

> > ### Comment · Reviewer_YXQf · 2025-11-26
> >
> > I would like to thank the authors for their response. However, my main reservations are still there. Also, after reading the other reviews and responses as well, I will keep my score.

---

### Official Review · Reviewer_LKJB · 2025-10-31

**Soundness:** 2
**Presentation:** 2
**Contribution:** 3
**Rating:** 2
**Confidence:** 3

**Summary:**

This work proposes a representation learning-based approach (CausalM) for node selection within branch-and-bound. Node selection policies govern the exploration of the branch-and-bound search tree by recommending relaxations to consider next at any stage of the solution process. Traditionally, MIP solvers employ policies based on simple greedy hueristics; this work belongs to a line of recent research that explores various deep ML approaches for node selection.

Fundamentally, the authors argue that prior ML-based approaches, which extract training data from solver traces, suffer from sampling bias because these traces tend to favor nodes with particular features. As a result, the authors frame node selection as a causal decision-making problem, where interventions (expanding a particular active node) influence the distribution of whether the selected node contains on optimal solution in the current solve state. The paper invokes causal reasoning mainly as a conceptual framing and motivation rather than through an explicit causal model, introducing an ML architecture that directly predicts the probability that a node contains an optimal solution as a proxy for causal estimation. The efficacy of the base approach, and an augmented approach that is more data-efficient, are both demonstrated via experiments on MIP benchmarks considered in prior work [Labassi et al. 2022].

**Strengths:**

1. The paper introduces a novel conceptual framing of node selection as a causal decision-making problem, directly addressing the bias that arises when relying on MIP solver traces for supervision signals in ML approaches to node selection.
2. Empirically, CausalM is a marked improvement over the chosen ML-based baselines for node selection, consistently outperforming these methods in at least one of solve time, number of problems solved to optimality, or generated tree size (and often all three).
3. The CausalM architecture and experimental setup are both clearly described in a high level of detail, enabling reproduction.

**Weaknesses:**

1. While a causal approach is well-motivated, the connection to the proposed representation learning architecture is unclear. Namely, this approach does not actually estimate the causal target (which would require defining a causal graph or structural equations).
2. There is no evidence provided (experimental or theoretical) to support the claim that the proposed ML approach achieves a lower confounding gap than prior correlation modeling approaches.
3. The experiments do not include a comparison to state-of-the-art RL-based baselines (e.g. Zhang et al. 2025), which generally outperform the supervised learning approach of Labassi et al (2022).

**Questions:**

Could the authors elaborate on how CausalM’s learning pipeline would not also be considered “correlation modeling?” Additionally, I would appreciate further clarification on the concerns raised in the weaknesses.

#### **Other comments**
* Theorem 1 is an immediate property of the branch-and-bound tree construction and would be better presented as an observation than a contribution.
* Page 1: “dominate” $\mapsto$ “dominant”
* Page 3: no space after “Recent work has also explored reinforcement learning”
* Page 4: it is not clear what part of equation (7) is coming from Kallus and Zhou (2021); the line reads as though their paper introduced TV distance.
* There is a significant amount of repeated information between tables 1/3 and 4/5.
* Given that there is no causal estimation, terminology like “causal linkage,” “causal signal,” “causal supervision,” etc. are ambiguous.

---

> ### Author Response · Authors · 2025-11-21
>
> **(1) Connection between causal framing and the representation learning architecture**
>
> We appreciate the reviewer’s concern and agree that our method does not perform explicit structural causal estimation. Our goal is not to recover structural equations but to reduce the influence of confounding by exploiting a causal perspective. Specifically, prior node-selection learning models estimate P(y | u_t = v, s_t) from solver traces, which entangles latent confounders z (e.g., depth, LP bounds) with the decision u_t. In contrast, our method introduces a representation-learning objective grounded in optimality transitivity, which defines an invariant causal relation between nodes and their ancestors that is independent of these confounders. Thus, while CausalM is implemented as a contrastive representation model, its supervision signal corresponds to an interventional view of node optimality.
>
> **(2) Evidence for mitigating the confounding gap**
>
> We acknowledge that the current version of the paper does not explicitly quantify the reduction in confounding bias. Our evaluation indirectly supports this claim: CausalM consistently improves generalization to unseen problem distributions and achieves smaller variance across instance types compared to correlation-based baselines (Tables 1–5). These results indicate that the learned representation is less dependent on solver-induced biases.
>
> **(3) Comparison with RL-based baselines**
>
> We thank the reviewer for this suggestion. We did not include Zhang et al.~(2025) in our comparisons because their RL approach relies on code that is not publicly available, which prevents us from reliably reproducing their results.

---

### Official Review · Reviewer_LopH · 2025-10-31

**Soundness:** 3
**Presentation:** 2
**Contribution:** 3
**Rating:** 6
**Confidence:** 2

**Summary:**

This paper studies the problem of learning a node selection rule for use in Branch and Bound solvers for mixed integer linear programs (MILPs).
Roughly speaking, these solvers construct a tree that recursively partitions the original MILP problem into subproblems.
Each node in this tree corresponds to a subproblem specified by an additional set of constraints, one introduced at each edge along the path to that node.
On each iteration, the BnB algorithm selects one of the nodes in the current tree to expand into new children nodes by imposing disjoint bounds on a fractional variable in the optimal solution to the LP relaxation at that node.
Traditional BnB implementations use hand-designed heuristics for node selection, but a recent line of work has explored the idea of learning node selection policies that perform well for a given distribution over problem instances.

This paper argues that most prior work on learning node selection rules only learn correlations between node features and selection quality and therefore fail to control for latent confounding variables.
Instead, motivated by causal modeling, this paper proposes a new approach to learning node selection policies that has a significantly different flavor from prior work: they learn a similarity function that takes as input a node and its ancestor where the learning objective is to make (node, ancestor) pairs that contain an optimal solution similar, and other pairs dissimilar.
The key observation that this technique exploits is the fact that if one node of the tree contains an optimal solution to the original MILP, then so do all of its ancestors.

The authors implement this approach using contrastive learning.
Each node is embedded by a bipartite graph encoder with learned parameters, and then the similarity between two nodes is measured by the cosine similarity of their embeddings.
The parameters of the encoder are learned via InfoNCE.

Finally, the authors conduct an extensive empirical evaluation of their proposed method and several baselines on several MILP benchmark and real-world datasets.

**Strengths:**

The problem of learned node selection rules in BnB MILP solvers is an interesting problem with many applications.
The proposed method is significantly different from the prior work and the experimental evaluation shows that it performs well.

**Weaknesses:**

I had a hard time understanding some of the core motivation of the paper, but this could be because I am relatively unfamiliar with causal analysis. I think the paper would be significantly improved by including additional discussion of why focusing on learning a similarity measure between nodes and their ancestors is consistent with a causality-motivated approach, while learning to predict scores from node features is not.

**Questions:**

1. In Section 3 the discussion revolves around the conditional distribution of $y$ given that node $v$ is selected at step $t$ when the search tree at step $t$ is equal to $s_t$. What is the joint distribution across these variables? I was guessing that there is a distribution over MILP instances which are then solved using a baseline BnB implementation, and the nodes present in the trees produced by that solver are the ones that appear in the distribution. There is some discussion of this in Appendix A.2.1, but I didn't quite follow. It would be helpful to be a bit more explicit about what this distribution corresponds to. I had trouble following precisely some of the theoretical motivation.
2. I don't quite understand the connection between optimality transitivity and causality. I agree that if one node contains an optimal solution then all of its ancestors must too. But it is less clear to me why this implies that learning a similarity function as proposed is accounting for latent confounding variables. It could be worth giving some intuition for the proof of Proposition 1.
3. What is the oracle baseline in the experiments? Is it the result of selecting nodes in BnB in such a way that the size of the tree is minimized?
4. It seems like when learning the similarity function, we might want to use the collection of all optimal solutions to the MILP when deciding the positive and negative examples. Does the data generation process use all optimal MILP solutions? If not, do we run the risk of learning bad similarity functions because we assign a label of 0 to some (node, ancestor) pairs where both contain an optimal MILP solution, just not one that we know of?

---

> ### Author Response · Authors · 2025-11-21
>
> **Q1. Clarification on Section 3 (Distribution and Confounding Variables)**
>
> We sincerely thank the reviewer for the careful reading. We would like to note that “the data sampling scheme” in Section 3 may have caused ambiguity. Our intention was not to describe bias in the data collection process, but rather the decision bias induced by the BnB solving procedure.
>
> As suggested, we have revised Section 3.1 in the updated manuscript to clarify this distinction. The updated text reads:
>
> “In practice, nodes that contain the optimal solution often share particular characteristics (e.g., those with small LP bounds, which are frequently expanded in solver runs).
> These characteristics arise because the B&B process progressively restricts the feasible region along the search path, causing optimal nodes to appear in parts of the tree where such confounder-related features are naturally present.
> Consequently, observational data obtained from solver runs inherently couples optimality with these confounders.
> When a correlation-based model is trained on this data, the estimated relationship P(y | u_t = v, s_t) can be dominated by these spurious associations, leading the model to rely on confounder-driven features rather than on information that truly reflects whether a node leads to the global optimum.”
>
> This refinement improves clarity but does not affect our theoretical results or empirical findings.
>
> **Q2. Connection between optimality transitivity and causality**
>
> We appreciate this insightful question. The link between optimality transitivity and causality lies in how the transitivity property defines a structural invariant that separates the causal mechanism of optimality from spurious correlations introduced by latent confounders.
>
> In correlation-based learning, the model estimates P(y | u_t = v, s_t), which is affected by latent variables such as node depth, LP bounds, or branching heuristics that influence both node features and selection decisions. The optimality transitivity property, stating that if a node contains an optimal solution, all its ancestors must also contain it, is invariant to these confounders because it depends solely on the feasibility regions induced by branching constraints. Learning a similarity function between node–ancestor pairs captures this invariant causal linkage: nodes sharing the same optimal solution are causally connected along the search tree, while others are not. The contrastive objective encourages alignment of causally related pairs and separation of unrelated ones, thereby implicitly controlling for unobserved confounders.
>
> **Q3. Definition of the Oracle baseline**
>
> The Oracle baseline is an idealized heuristic that always expands a node containing the optimal solution, simulating a perfect expert with knowledge of the optimum. However, it does not guarantee the smallest possible BnB tree, since multiple nodes may contain the optimal solution at a given depth. It thus serves as an upper-bound reference rather than a minimal-tree benchmark.
>
> We have revised the definition of the Oracle baseline in Section A.4.2:
>
> “Oracle: an expert diving policy constructed by first solving the MILP to obtain the optimal solution and then, during the BnB process, always selecting the active node whose feasible region contains this optimal solution.”
>
> **Q4. Use of all optimal solutions in data generation**
>
> We thank the reviewer for the thoughtful comment. We agree that some MILP instances may contain multiple optimal solutions, whereas our current dataset construction uses a single optimal path per instance.
>
> We have now explicitly noted this limitation and discuss potential extensions to include multiple optimal paths in the Conclusion section.

---

### Meta-Review · Area_Chair_i47Q · 2026-01-06

**Summary:**

**Summary:**
This paper proposes causal, a method for learning node selection policies in branch-and-bound solvers for mixed integer linear programs. The core insight is that one can use contrastive learning with optimality transitivity to provide valuable supervision signals. The authors argue this approach captures causal relationships, thereby avoiding spurious patterns in solver traces.

**Rationale:**
The paper presents an interesting application of contrastive learning to branch and bound node selection. The empirical results are suggestive, however the theoretical results appear to be relatively week and the causal framing beyond motivation appears weak as well. As pointed out by others, this can be viewed as learning invariant embeddings along optimal paths. Further, the missing modern baselines and small datasets make it hard to argue that this work clears the ICLR bar on empirical grounds.

**Reviewer Concerns:**

loph:
- True causality: The authors tried to address this framing but unconvincingly.
- Theoretical motivations: The reviewer indicated that section 3 was hard to follow, and the connection to pearl's do-calculus was hard to follow. The authors largely dropped this point.

lkjb:
- True causality: The authors tried to address this framing but unconvincingly. This reviewer noted that the authors don't actually estimate a causal target, making the causal framing more challenging to defend.

yxqf:
- True causality: The authors tried to address this framing but unconvincingly.
- limited exerpiments: The authors cited lack of available code in RL based baselines but the broader concern about benchmark size and baselines remains.

dugo:
- limited exerpiments: The authors cited lack of available code in RL based baselines but the broader concern about benchmark size and baselines remains.
- Theoretical motivations: The reviewer indicated that section 3 was hard to follow, and the connection to pearl's do-calculus was hard to follow. The authors largely dropped this point.

**Reviewer Scores:**

- loph, 6 -> 6, the questions were answered but the reviewer wasn't enthusiastic
- lkjb, 2 -> 2, the core concerns remain
- yxqf, 4 -> 4, the reviewer explicitly moved to maintain their scores and this appears to be because their core concerns remain as well
- dugo, 4 -> 5, some concerns were addressed
- u9cw, n/a

---

### Decision · Program_Chairs · 2026-01-26

Reject